# Facilitating Antiviral Drug Discovery Using Genetic and Evolutionary Knowledge

**DOI:** 10.3390/v13112117

**Published:** 2021-10-20

**Authors:** Xuan Xu, Qing-Ye Zhang, Xin-Yi Chu, Yuan Quan, Bo-Min Lv, Hong-Yu Zhang

**Affiliations:** Hubei Key Laboratory of Agricultural Bioinformatics, College of Informatics, Huazhong Agricultural University, Wuhan 430070, China; xxpinky@webmail.hzau.edu.cn (X.X.); chuxy@webmail.hzau.edu.cn (X.-Y.C.); quanyuan@mail.hzau.edu.cn (Y.Q.); lbm612@webmail.hzau.edu.cn (B.-M.L.)

**Keywords:** genetics, evolution, host receptors

## Abstract

Over the course of human history, billions of people worldwide have been infected by various viruses. Despite rapid progress in the development of biomedical techniques, it is still a significant challenge to find promising new antiviral targets and drugs. In the past, antiviral drugs mainly targeted viral proteins when they were used as part of treatment strategies. Since the virus mutation rate is much faster than that of the host, such drugs feature drug resistance and narrow-spectrum antiviral problems. Therefore, the targeting of host molecules has gradually become an important area of research for the development of antiviral drugs. In recent years, rapid advances in high-throughput sequencing techniques have enabled numerous genetic studies (such as genome-wide association studies (GWAS), clustered regularly interspersed short palindromic repeats (CRISPR) screening, etc.) for human diseases, providing valuable genetic and evolutionary resources. Furthermore, it has been revealed that successful drug targets exhibit similar genetic and evolutionary features, which are of great value in identifying promising drug targets and discovering new drugs. Considering these developments, in this article the authors propose a host-targeted antiviral drug discovery strategy based on knowledge of genetics and evolution. We first comprehensively summarized the genetic, subcellular location, and evolutionary features of the human genes that have been successfully used as antiviral targets. Next, the summarized features were used to screen novel druggable antiviral targets and to find potential antiviral drugs, in an attempt to promote the discovery of new antiviral drugs.

## 1. Introduction

Since the earliest yellow fever virus was discovered in 1901, the number of viruses has continued to increase. On average, three to four new species are found each year. So far, more than two hundred viruses that infect humans have been identified [1]. Human viral infection is the most unpredictable disease among infectious diseases. Currently, there are several viruses with high mortality rates in the world. For instance, Coronavirus disease (COVID-19) caused by SARS-CoV-2 has led to a global pandemic and can result in a series of respiratory diseases, such as pneumonia and lung failure [2,3,4]; Ebola virus causes a range of highly pathogenic symptoms, such as hemorrhagic fever, and is highly contagious, with a mortality rate of 57%–90% [5]; Severe Acute Respiratory Syndrome (SARS), has affected more than 8000 people and killed 774 people by July 2003 [6]; Epstein-Barr virus (EBV), a common human herpes DNA virus, can lead to lifelong infection in more than 90 percent of the population, and causes a variety of human malignancies [7]; Human Immunodeficiency Viruses (HIV/AIDS) causes severe defects in human cellular immune function. As of 2018, 77.3 million people worldwide have been infected with HIV, and 35.4 million people have died from AIDS-related diseases [8]. Thousands of human plagues are still found every year worldwide [9]. These viruses usually have the characteristics of wide spread and strong concealment. As a result, mortality from viral diseases remains high. Due to the limitation of virus resistance, effective preventive measures and treatment drugs are still lacking for many viral diseases. There is a pressing need for the development of antiviral drugs.

Antiviral drugs are usually classified according to their effects: anti-HIV drugs (such as Zindovudine, Abacavir, Nelfinavir, Delavirdine, Nevirapine, etc.) [10,11]; anti-cytomegalovirus (CMV) drugs (such as Ganciclovir, Valganciclovir, Cidofovir, Formivirsen, etc.) [12,13]; anti-hepatitis virus drugs (such as Telaprevir, Ribavirin, Simeprevir, Boceprevir, Sofosbuvir, etc.) [14,15]; anti-herpes virus drugs (such as Acyclovir, Valacilovir, Idoxuridine, Brivudin, etc.) [16,17]; and anti-influenza and respiratory virus drugs (such as Amantadine, Rimantadine, Osehamivir, Zanamivir, etc.) [18]. Although there are dozens of drugs available for the treatment of some important viral diseases, they only target a small number of viral pathogens [19,20,21]. In addition, the occasional appearance of more pathogenic strains of known or previously unknown viruses continues to raise public health concerns and reminds people of the need for effective treatments. The research statistics on antiviral drugs in the NCBI PubMed from 1945 to 2021 showed that the number of research papers dedicated to this topic has increased year by year, from only one in 1945 to a maximum of 17,198 in 2013, indicating a growing interest in the development of antiviral drugs.

Drug development has always been a very long and costly process, whose comprehensive cost estimates range from hundreds of thousands of dollars to approximately $2.6 billion [22,23,24]. Among thousands of new chemical structures, only a few prove to be potential drugs [22]. It is estimated that only 10.4% of new phase I clinical development projects were finally approved by the US Food and Drug Administration (FDA) between 2003 and 2011 [25]. Some studies have shown that the efficiency of research and development (R&D) has been steadily declining, as measured by the number of new drugs that are put on the market per billion US dollars of Research and Development expenditure in the global biotechnology and pharmaceutical industries [23,26]. One main reason for failed drug development is the lack of effectiveness of drug targets [27,28]. Considering the fact that viruses lack a cell structure, their genetic material is easily perturbed by the external environment and the host cellular molecular environment. Traditional antiviral drugs mostly inhibit the enzymes that are essential for virus reproduction at different stages of virus replication, thereby blocking this replication [19,29]. However, more and more studies have shown that this pathogen-targeting strategy, though successful in many cases, is not effective enough to combat the emergence of drug resistance [30].

Over the past few decades, significant advances in high-throughput sequencing, analytical techniques, and molecular biology, have brought deeper understanding of the genetic and evolutionary mechanisms of viral infection and provided new opportunities for therapeutic interventions [31]. At present, a large number of gene loci related to susceptibility to viral diseases (such as HIV, hepatitis, dengue (DEN) etc.) have been discovered by genome-wide association studies (GWAS) [32]. These results further elucidate the genetic structure of susceptibility to infectious diseases. It is known that genetic diversity drives evolution and contributes to adaptation to new environments. Gelbar et al. used ultra-deep sequencing to examine 43 clinical samples taken from early human infections with HIV, respiratory syncytial virus pneumonia (RSV), and Cytomegalovirus (CMV). They identified the presence of multiple distinct genotypes in the HIV and CMV samples as the major drivers of increased diversity [33]. Pairo-Castineira et al. performed a GWAS comparing 2244 critically ill COVID-19 patients with heathy individuals in the UK. Subsequently, using the transcriptome-wide Mendelian randomization (MR) method to investigate the causal relationship between potentially modifiable risk factors and health outcomes, they found that the high expression of encoding tyrosine kinase 2 (TYK2) was significantly related to critical COVID-19. Therefore, the authors speculated that the TYK2-targeted drug (JAK inhibitor Baricitinib) may have the potential to combat SARS-CoV-2 [34]. Meanwhile, host-targeted antiviral drug discovery strategies have become popular in recent years. Since the mutation rate of the host protein is significantly lower than that of the virus, host-targeted antiviral drugs are thought to offer a higher genetic barrier to against mutations than direct-acting antiviral drugs [35]. Ullah et al. recently discovered that the inhibitor of human RACK1 protein shows similar efficacy to the approved anti-herpes drug acyclovir, and that RACK1 protein is a potential target for the development of broad-spectrum antiviral drugs [36]. Tyrrell et al. selected the endoplasmic reticulum a-glucosidases as a target based on the glycosylation mechanism, and iminosugars were shown to have potential efficacy in difficult-to-treat infections [37]. Up to now, this strategy has also been successful in the treatment of DEN [38] and other viral diseases [39,40,41]. Burmeister et al. proposed a treatment method based on ecological evolution systems. The virus microorganisms and the host’s internal environment are regarded as an ecosystem, and host and microbial pathogens are part of this complex interaction system [42]. It can be seen that the strategy of targeting host factors circumvents the barrier created by the high-frequency mutation of viruses, and provides a unique opportunity for the development of more broad-spectrum antiviral drugs [42].

On the other hand, recent studies have also found that successful drug targets often have common genetic and evolutionary features, which are of great help in screening potential drug targets and discovering new drugs [43,44,45]. Inspired by the above information, in this study we first summarized the genetic characteristics, cellular sublocation, and evolutionary features of existing approved host-targeted antiviral drug targets. Next, based on the summarized features, potential host-aimed antiviral targets were screened, in an attempt to promote the development of new antiviral drugs.

## 2. Feature Analysis for Host Targets of Approved Antiviral Drugs

### 2.1. Genetic Features

For viruses, the primary barrier to invasion is the cell’s innate immunity. This antiviral response first detects viral components through host cells, then transduces the antiviral signals, transcription and translation of antiviral effectors, and finally establishes the body’s antiviral state [46]. Combined with the host-targeted strategy, much progress has been made in identifying specific human genetic variations that help to increase sensitivity or resistance to viral diseases [47]. Currently, the most popular genetic testing methods focus on the discovery of a disease’s potentially susceptible genes through GWAS [48,49,50] and the detection of genetic polymorphisms related to the phenotype of specific viral diseases through whole-exome sequencing (WES) [51,52]. For example, Akcay et al. recently reported that HLA-DP/DQ mutations contribute to HBV persistence and resistance to vaccines. Besides, susceptible alleles in the HLA-DP/DQ locus are more common in Asian populations than in Whites, which may explain why chronic HBV infection rates are higher in Asian populations [50]. Additionally, Hashemi et al. declared that host-targeted antiviral drugs may be subject to host gene polymorphisms that alter their ability to inhibit the function of target proteins [53]. Indeed, in the clinical setting, suboptimal response to Alisporivir has been reported in 10–15% of patients, which may be due to the potential influence of single nucleotide polymorphisms (SNPs) [54]. Another study tested whole exon mutations at the individual level and showed that the progression of HIV-1 disease is related to microbial infection and innate immune activation, and the role of microbes in the damaged intestinal mucosa is thought to be related to disease progression. Therefore, microbial translocation and the activation of innate pattern recognition receptors (PPR) may affect disease progression and non-AIDS complications [55]. Gene mutations in EC 004 (*TAB2*), LTNP 005 (*PIK3C2B*), LTNP 007 (*FGD6*), and LTNP 008 (*PRKDC*) are also particularly significant for the slow-progressing phenotype of HIV [55]. These genes, as well as the immunological and cellular biological mechanisms involved in them, could be the subject of further research on the pathogenesis and disease progression of HIV, and may also become potential new drug targets. In addition, based on a genome-wide loss-of-function CRISPR screen, Daniloski et al. obtained highly ranked genes in host. The dysfunction of these genes significantly reduces the infection rate of SARS-CoV-2. Moreover, nine of these genes are druggable genes corresponding to 26 small molecule inhibitors. The experimental results showed that seven of the inhibitors can reduce the SARS-CoV-2 viral load by more than 100-fold. Interestingly, among these seven effective antiviral agents, four target the same gene, *PI3K catalytic subunit type 3* (*PI3KC3*) [56]. Therefore, the authors contended that CRISPR is an effective genetic method for screening antiviral targets [56,57,58]. Furthermore, since GWAS can only identify genes that are significantly associated with the disease, there is an increasing number of studies incorporating MR to identify the causal relationship between gene mutations and disease. By combining GWAS and MR methods, some host genes, such as *IFNAR2*, *TYK2*, *JAK1*, *JAK2*, *TNF*, *IL6*, *TMPRSS2*, and *ILIR1* have been identified as potential therapeutic targets for COVID-19 [34,59]. These findings provide important clues for a better understanding of the mechanism of COVID-19. Taken together, genetics-identified disease genes are considered a promising source of drug targets to help increase the success rate of antivirus drug development [28].

In a previous study [28], we combined the information from eight disease gene databases (including Clinvar [60], Online Mendelian Inheritance in Man (OMIM) [61], The Human Gene Mutation Database (HGMD) [62], Orphanet [63], GWASdb [64], INtegrated TaRget gEne PredItion (INTREPID) [65], Genetic Association Database (GAD) [66], and DisGeNET [67]) and used the degree of clinical support to award these genes different quality scores (termed druggability score). Through a comprehensive analysis, it was confirmed that these disease genetics-derived druggability scores can be used to measure the association between genes and diseases, which offers important values for screening potential targets and predicting drug activity [28]. The higher the score, the stronger the association between the gene and the disease, and the easier it is to develop the gene as a drug target.

To analyze the genetic features of antiviral drug targets, firstly, 36 known host targets of approved antiviral drugs were downloaded from DrugBank [68] (Appendix A). Based on the druggability scores derived from the SCG-Drug database (Quan et al. 2019), 32 targets had relevant score information (accounting for 88.9% of the total 36 targets). Next, the median score of these targets with 34 viral diseases was calculated to measure the strength of the association between genes and pathogenic viruses. The results showed that 13 genes received genetic scores equal to or greater than three. Besides, 27 targets were documented in association with more than one viral disease, as shown in the last column of Table 1. 

### 2.2. Subcellular Location Features

The coding capacity of the viral DNA genome is limited, so it is necessary to use host cell factors to promote the replication of its genome and to produce progeny viruses [69]. A critical initial step in the life cycle of an infectious virus is the identification of and interaction with the host proteins. Therefore, to initiate this cycle, the viral genome needs to be transported from viral particles to the cytoplasm through different viral attachment proteins on their shells [70]. The particles are first attached to the specific proteins on the surface of the host cell. These specific proteins are called receptors. The presence of receptors on the cell surface is a major determinant of viral host selectivity and plays a key regulatory role in host range and viral pathogenesis [71]. Until now, antiviral drug strategies targeting cell surface receptors have made progress in the treatment of different viral infectious diseases [72]. One of the main areas of focus for research is anti-HIV drugs. HIV entry into target cells is a complex process, consisting of multiple biological responses, which is triggered by the interaction between the HIV envelope protein (Env) glycoprotein gp120 and the host cell receptor *CD4*. This interaction causes a conformational change in gp120, exposing a co-receptor binding site [73]. Subsequently, after binding to the *CCR5* or *CXCR4* core receptor of the primary HIV isolate, Env undergoes further conformational changes, gp41-mediated membrane fusion, and virion core entry into the cytoplasm [74]. Therefore, the polymorphism of the *CCR5* gene can affect the spread of HIV or the development of the disease. Targeting the process of HIV invasion into cells is considered to be an effective method to inhibit HIV replication [11]. The anti-HIV efficacy of small molecule *CCR5* antagonists, such as Cenicriviroc [75] and Vicriviroc (SCH-D) [76], has proven this theory.

Given the above findings, a subcellular localization feature needs to be considered in drug development and protein function development. By analyzing the mechanism of virus invasion, it can be speculated that cell membrane receptors are more likely to become potential targets for antiviral drugs. To test this conjecture, we also began with the 36 known antiviral human targets. The Uniprot database was used to retrieve their subcellular localization information [77]. To improve the accuracy and credibility of the results, the manually annotated and reviewed Swiss-Prot database was chosen for the analysis [77]. A total of 20,233 pieces of human genetic information was collected from the Swiss–Prot database, of which 14,574 were labeled with subcellular localization information. Subcellular location information was available on the database for 32 out of the 36 known antiviral human targets, and 21 were located on the cell membrane (65.6%). It was observed that the antiviral targets were indeed significantly enriched on the cell membrane surface (*p* < 0.001, hypergeometric test). This characteristic can be used as a key indicator for subsequent antiviral host target screening. It is worth noting that among the 20 scored cell membrane receptors, 11 received a median SCG score of three, accounting for 84.6% (11/13) of the total number of targets with a score of three or more. Therefore, the cell membrane receptors demonstrated a high genetic association with viral diseases.

### 2.3. Evolutionary Features

Viruses initially interact with the host through their capsid or envelope proteins. There is an argument that the primitive capsid protein-encoding genes are recruited from cellular hosts, which implies that the coevolution between virus and host is deeply rooted in the origins of viruses [78]. Furthermore, some studies have shown that the evolution of the virus and parasite host range play an important role in the emergence of infectious diseases [78,79]. Coevolution between viruses and hosts has also led to larger genetic differences between replication populations, which are often related to the range of hosts that co-evolutionary viruses can infect [80,81]. For example, health disasters caused by influenza viruses have been common in human history. One of the causes of influenza virus epidemics is its ability to continuously perform antigen transformation through genetic diversity. At the same time, the host’s immune defense mechanism is constantly evolving. This process might be compared with the evolutionary race between predators and prey [78,82]. By studying the mechanism of host resistance to infection, SNPs of the interferon-inducible transmembrane proteins (IFITMs), especially *ifitm3* gene, were found to be able to increase or decrease susceptibility to influenza infection [83,84], showing the antiviral significance of the host protein’s evolution. Synergistic benefits can also be produced by integrating the virus-host evolutionary mechanism and the interaction between pathogens and hosts to improve therapeutic effects [42]. So far, two effective treatment strategies, evolutionary trade-offs and/or competitive effects, have been proposed, based on the evolutionary model in which viruses in the host control or eliminate pathogens [42,85,86]. Furthermore, Phillips et al. found that the ability of host chaperone proteins to influence virus evolution is a key factor in determining viral protein mutation fitness, which provides new insights for the design of antiviral treatments [87].

Tomislav et al. showed that the pathogenicity of disease genes is closely related to their evolutionary origin [88]. The results of research by Wang et al. indicated that successful targets share some common evolutionary characteristics, and suggested that evolutionary information can help to identify the drug targets with the greatest potential for therapeutic development [43,45]. An in-depth understanding of the antiviral activity and action mechanism of host target genes will help to develop broad-spectrum antiviral drugs that enhance their activity or have similar effects [89].

There are more than 30 popular algorithms through which to infer the origins of gene families [90,91,92]. It is a challenge to obtain accurate gene ages because the age of the gene families depends on the accuracy of the genetic algorithm. To solve this problem, Liebeskind et. al. provided a relatively consistent genetic age data set and mapped the species with orthologous genomes to a reference tree from SwissTree to infer the age of each gene [93]. Based on the rules, human genes are divided into the following eight classes by origin: (1) cellular organism, (2) the common ancestors of eukaryotes and archaea (Euk_Archaea), (3) horizontal gene transfer of bacteria (Euk + Bac), (4) Eukaryota, (5) Opisthokonta, (6) Eumetazoa, (7) vertebrata, (8) mammals. It is believed that this set of data includes minor errors.

Based on the above knowledge, the evolutionary origin of 36 human antiviral targets were analyzed. The UniprotKB data corresponding to 36 targets were obtained by querying the UniprotKB database. The evolution information statistics results indicated that the 36 UniProtKB were mainly distributed in genes originating in class 6 (Eumetazoa) (14/36, *p* = 0.04, hypergeometric test) (shown in Figure 1a). As for the 21 cell membrane receptors, 13 of them originated in the Eumetazoan period, and the trend of enrichment was more obvious (*p* = 0.00034, hypergeometric test) (shown as in Figure 1b). The results of the gene functional annotation using DAVID showed that these receptors are significantly related to immune response (*p* =1.4 × 10^−8^).

A previous evolutionary study of 658 successful targets by Quan et al. indicated that these genes are significantly enriched in cellular organism, euk + bac, and Eumetazoa [45]. Our results were partially consistent with their study, suggesting that human genes originating in the Eumetazoan period have greater potential to become antiviral targets.

## 3. Rational Screening of Potential Antiviral Host Targets

It is often inadequate to screen targets based solely on a single feature summarized by omics data because they usually depend on large-scale experiments and cannot comprehensively explain the causal relationship between human diseases [45,94]. Besides, biotechnology continues to identify an increasing number of disease-related genes. Not all of them are potential drug targets. Identifying new and effective targets remains a major priority for modern drug discovery [95]. Based on the analysis of the genetic features, subcellular localization, and evolutionary features of approved antiviral host targets, it is summarized that an effective antiviral target generally has the following three characteristics: (1) a median of genetic score(s) associated with viral disease(s) equal to or greater than three; (2) subcellular localization on the cell membrane; (3) origins in the Eumetazoan period. By using these criteria, we screened out some potential antiviral host targets (Figure 2).

### 3.1. Target Screening by Summarized Features

As mentioned above, the SCG-Drug database integrates information from eight common genetic disease databases and provides different druggability scores for these genes. It provides us with a huge data set for screening potential antiviral drug targets. We started with 914,190 gene-disease associations in the SCG-Drug database and obtained 19,232 genes by removing repetitions. These genes were used for screening according to the three features described above. First, through genetic screening, 7214 genes were found to be associated with 34 viral diseases. For each gene, the median score associated with viral disease(s) was taken as its genetic score. Next, these genes were divided into two categories based on whether there was relevant information on the three well-known drug databases (Therapeutic Target Database (TTD) [96], DGIdb [97], and DrugBank [68]). Of the 2277 genes with relevant drug information on the databases, we found that 687 are located on the cell membrane and 248 originated in the Eumetazoan period. For the 4937 genes with no drug information, 744 were found to be located on the cell membrane, and 239 originated in the Eumetazoan period. To find candidate genes that are strongly associated with viral disease(s), genes with genetic scores equal to or greater than three were selected. Furthermore, through searching the studies available on NCBI PubMed database, we selected genes with virus-related reports as candidate targets. Finally, 35 candidate targets that met all of the above requirements were obtained (Table 2). For the 13 candidate targets with drug information, their corresponding drugs may be able to treat viral diseases through drug repositioning. As for the remaining 22 candidate targets without drug information, novel antiviral drugs may be derived from them through structure-based drug design.

### 3.2. Target Screening by Protein Structures

Although we can use various technologies to obtain a large number of potential therapeutic targets, it remains a great challenge to bring these targets into clinical trials. The reported statistics results showed that about 10% of genes in the entire human genome are involved in the development of disease, that is, about 3000 potential targets are suitable for drug treatment [98]. Therefore, predicting whether a protein features a pocket that can bind drug-like molecules with high affinity (druggability [99]) is of great significance in the target screening stage of drug discovery [100]. In order to improve the efficiency of target discovery, many computational methods have been developed to predict target druggability [101,102,103,104]. PockDrug [101] is a reliable model for predicting the druggability of pockets, which is based on a linear discriminant analysis of 52 molecular descriptors. It selects the most stable and effective model according to different prediction objects. In addition, PockDrug maintained the best combination of molecular descriptors that affect the pharmacological properties of the pocket. With the best combination, PockDrug can achieve an average accuracy of about 87% [101].

For those targets about which there is no drug information, we need to perform druggability analysis based on protein structures to determine whether it has the potential to be a small molecule drug target. The PDB structure of candidate targets were queried in Swiss-Prot. However, not all of these candidate genes feature known protein structures. To improve the accuracy of the results, we only performed the druggability prediction on the potential targets with reported crystal structures. Therefore, 12 candidate genes with known protein crystal structures were selected for druggability analysis. All the predicted information by PockDrug is summarized in Table 3 and Appendix A. Pockets with druggability scores of 0.5 or more were considered druggable.

We used the “predict druggability with proteins” function. Next, we uploaded the corresponding PDB file, selected the Fpocket estimation method (not guided by ligand information), and set the Ligand proximity threshold to the default, 5.5 Å. The complete and detailed prediction results are shown in Appendix A.

## 4. Antiviral Drug Discovery Based on Screened Host Targets

### 4.1. Drug Repositioning

Drug repositioning has always been an important way to reduce drug development costs and speed up the research process [106]. At present, the number of approved antiviral drugs targeting host genes is significantly higher than that of drugs directly targeting viral genes [107]. Therefore, it is feasible to adopt a host-centered strategy and to integrate genetic and evolutionary knowledge to increase the success rate of drug repositioning. Among the 35 candidate targets listed in Table 2, 13 targets can be found with relevant approved drug information in the DrugBank database. It is believed that the existing approved drugs targeting these genes may be used in antiviral therapies through drug repositioning.

By searching the ClinicalTrials databases, the efficacy of some repositioned drugs was supported by the clinical evidence. For instance, *IL1R1* (Interleukin-1 receptor type 1) encodes a cytokine receptor belonging to the interleukin-1 receptor family, which participates in the regulation of innate immune and inflammatory processes [108]. Anakinra is a human interleukin-1 receptor antagonist approved for the treatment of adult rheumatoid arthritis (RA) and neonatal multiple inflammatory diseases (NOMID). It was predicted by disease genetics that drugs targeting *IL1R1* would be used for Paramyxoviridae infections, such as respiratory syncytial virus infections, influenza, and so on (Table 2, Appendix A). According to the records of the ClinicalTrials database, Anakinra is currently being studied for the treatment of COVID-19. Related ClinicalTrials.gov identifiers are NCT04362943, NCT04364009, NCT04330638, NCT02735707, NCT04362111, NCT04339712, NCT04366232, NCT04341584, etc. Some of these studies are already in phase 3 or 4 (NCT02735707, NCT04330638, NCT04364009, NCT04362111).

The gene *AGTR1* (Type-1 angiotensin II receptor) is one of the important effect factors in mediating the volume and blood pressure of the cardiovascular system [109]. The genetic data suggest that drugs targeting *AGTR1* can be used in the treatment of viral diseases, such as Paramyxoviridae infections, AIDS dementia complex, etc. (Table 2, Appendix A). Losartan is an angiotensin II receptor blocker used to treat high blood pressure and diabetic nephropathy and to reduce the risk of stroke. Losartan is indeed being investigated for the treatment of respiratory syndrome / coronavirus infection (NCT04312009, NCT04311177, NCT04340557) and HIV (NCT01852942, NCT01529749), which partially coincides with our predictions.

Likewise, Formoterol, targeting *ADRB2* (Beta-2 adrenergic receptor), *etc*., is an inhaled beta-2 agonist that was approved by the FDA in 2001 for the treatment of chronic obstructive pulmonary disease (COPD) and asthma [110]. It was inferred through genetic data that it can be used in the treatment of paramyxoviridae infections, respiratory syncytial virus infections, etc. (Table 2, Appendix A). Indeed, there is a phase 3 study that uses it for COVID-19 treatment (NCT04331054).

### 4.2. De Novo Drug Discovery

It can be seen from the results of the druggability prediction (Table 3, Appendix A) that candidate targets including *FAS, CD209, LGR5, CRHR2, AGER, BIN1, BSG, IL1RL1, PARD6A, RGS7, RHOU*, and *TNFSF10* have one or more druggable pockets. These twelve potential antiviral host target genes are divided into two types, according to whether they are known targets or not. One is the recognized target type, including *CD209*, *FAS*, *LGR5*, and *CRHR2* genes (however, without drug information); the other is the non-recognized type, including *AGER*, *BIN1*, *BSG*, *IL1RL1*, *PARD6A*, *RGS7*, *RHOU*, and *TNFSF10* genes. The twelve potential antiviral host targets have no approved drug information, therefore they may be used for de novo drug discovery.

The *CD209* gene encodes the pathogen-recognition receptor, *CD209* antigen, which is expressed on the surface of immature dendritic cells and is involved in the initiation of the primary immune response [111]. Many reports have shown that the *CD209* antigen could act as an attachment receptor for Ebolavirus [112], HIV-1 and HIV-2 [112], Dengue virus [113], and others. The encoding protein’s crystal structure has been released (PDB id: 2XR6), and the predicted best druggable pocket is shown in Figure 3a. Based on the identified cavity, more novel drug screening work is worth carrying out.

The *LGR5* gene encodes the Leucine-rich repeat-containing G-protein coupled receptor 5 for R-spondins that potentiates the canonical Wnt signaling pathway [114]. The *LGR5* gene is a proven target and has been used as a marker of adult tissue stem cells in the intestine, stomach, hair follicle, and mammary epithelium [115]. The receptor protein’s crystal structure was released and deposited in the Protein Data Bank (PDB id: 4ufr), and the predicted best druggable pocket was the cyan surface, as shown in Figure 3b. The pocket could be used as an important cavity for drug screening.

The *CRHR2* gene encodes the corticotropin-releasing factor receptor 2. The protein’s crystal structure was released and deposited in the Protein Data Bank (PDB id: 3n93). Its molecular functions include corticotropin-releasing factor receptor activity, corticotropin-releasing hormone receptor activity, and peptide hormone binding [116]. The predicted best druggable pocket was the cyan surface, as shown in Figure 3c.

The *FAS* gene belongs to the tumor necrosis factor receptor superfamily, and encodes the receptor for *TNFSF6*/*FASLG* [117]. The molecular functions of *FAS* include calmodulin binding, identical protein binding, kinase binding, signaling receptor activity, and transmembrane signaling receptor activity [118,119]. The *FAS* gene is a proven target and is involved in autoimmune lymphoproliferative syndrome 1A disease. The encoding protein’s crystal structure has been released (PDB id: 3tje), and the predicted best druggable pocket is shown in Figure 3d. Follow-up drug screening can be carried out based on this identified cavity.

The *AGER* gene encodes the advanced glycosylation end-product-specific receptor that mediates interactions between advanced glycosylation end products (AGE) and acts as a mediator of both acute and chronic vascular inflammation [120]. *AGER* is a member of the immunoglobulin superfamily of cell surface molecules, and *AGE/AGER* interaction has been linked to the regulation of the production/expression of TNF-alpha, oxidative stress, cancer, and endothelial dysfunction in type 2 diabetes [121]. The encoding protein’s crystal structure has been released (PDB id: 3o3u), and the predicted best druggable pocket is shown in Figure 3e. Based on the identified cavity, more structure-based virtual screening could be performed in the future.

The *BIN1* gene encodes the Myc box-dependent-interacting protein 1 that is a membrane deforming protein. It was found to be a tumor repressor due to its interaction with MYC oncoproteins [122]. The *BIN1* gene is also involved in DNA repair, cell cycle progression, cytoskeleton regulation, apoptosis, and regulating cellular *BACE1* levels [123]. The protein of the *BIN1* gene’s crystal structure was released and deposited in the Protein Data Bank (PDB id 2fic). Based on the crystal structure, the best druggable pocket was predicted and appeared in the cyan surface (Figure 3f).

The *IL1RL1* gene belongs to the interleukin-1 receptor family of cytokines and encodes the interleukin-1 receptor-like 1 protein, which is the receptor for interleukin-33 (IL-33) and whose signaling requires the association of the coreceptor *IL1RAP* [124]. The protein’s alternative name is the ST2 protein. The signaling of *IL-33* is generated by its ligand-binding primary receptor, ST2, and the subsequent recruitment of accessory receptor IL-1RAP, which results in the juxtaposition of intracellular toll/interleukin-1 receptor domains of both receptors, which are necessary and sufficient for the activation of the NF-κB and MAPK pathways in the target cells [125]. The ST2 protein’s crystal structure has been released (PDB id: 4kc3), and the predicted best druggable pocket is shown in Figure 3g. This receptor was also described as a negative regulator of Toll-like receptor-IL-1 receptor signaling, and it is worthy of further study as a potential antiviral target.

The *PARD6A* gene encodes the partitioning defective 6 homolog alpha protein. Its short name is PAR-6, PAR6-alpha, or PAR-6A; the protein’s alternative name is PAR6C, or Tax interaction protein 40 (TIP-40). PAR-6 protein is involved in asymmetrical cell division, cell polarization, and cell transformation processes [126]. The crystal structure of PAR-6 protein has been released and deposited in the database (PDB id: 1wmh), and the predicted best druggable pocket is shown in Figure 3h.

The *RGS7* gene encodes the regulator of G-protein signaling 7, which belongs to the regulator of the G-protein signaling superfamily. The combination of RGS7 protein and Gbeta5 constitutes a crucial regulator of G protein-coupled receptor signaling in the visual and nervous systems [127]. *RGS7* also plays an important role in synaptic vesicle exocytosis through its interaction with snapin [128]. The RGS7 protein’s crystal structure (PDB id:2a72) and predicted best druggable pocket is shown in Figure 3i. RGS7 protein also modulates the activity of potassium channels [127]. As a potential antiviral target, it offers significant research potential.

The *RHOU* gene encodes the Rho-related GTP-binding protein, RhoU. The protein’s alternative name is CDC42-like GTPase 1, GTP-binding protein-like 1, Wnt-1 responsive Cdc42 homolog 1 (Wrch-1), or Ryu GTPase. Active Wrch-1 can stimulate quiescent cells to reenter the cell cycle. Wrch-1 could regulate the actin cytoskeleton, cell morphology, cell proliferation, and migration [129]. The Wrch-1 protein’s crystal structure has been determined and deposited in the database (PDB id: 2q3h). The predicted best druggable pocket is shown in Figure 3j. Antiviral drug screening could be performed based on this potential antiviral host target.

The *TNFSF10* gene encodes the protein that belongs to tumor necrosis factor ligand superfamily member 10. The protein’s crystal structure was released and deposited in the Protein Data Bank (PDB id: 1dg6). The molecular functions include cytokine activity, identical protein binding, signaling receptor binding, tumor necrosis factor receptor binding, and others [130,131]. The predicted best druggable pocket is the cyan surface shown in Figure 3k. The pocket could be used as a valid cavity for drug screening.

## 5. Discussion and Conclusions

In recent years, new antiviral drugs have continued to appear, but the incidence of viral infections has continued to increase. Genetic and evolutionary mechanisms underlie the outbreak and epidemic of viruses. In particular, the continuous outbreak of high-mortality viruses, such as COVID-19, HIV, Ebola virus, DEN, and influenza virus, has created an urgent need for research into and development of antiviral drugs. In general, the challenge for antiviral drug discovery arises from the mutation and drug-resistance of viruses. For instance, most antibodies can be evaded by single mutations of the influenza virus, which leads to simultaneous outbreaks of multiple influenza virus subtypes [132]. Traditional antiviral drugs are prone to developing drug resistance after long-term use due to this characteristic of the virus. Host-targeting antivirals not only provide a higher genetic barrier to drug resistance, but also exhibit broad-spectrum antiviral activities and are likely to be effective against newly emerging viruses [11]. Currently, there are multiple avenues under investigation as therapeutic strategies for host-targeted antivirals, such as certain innate or adaptive immune host activities that respond to viral infections (i.e., TLR agonists [133], CCR5 antagonists [75,134,135], etc.) and the regulation of inflammatory pathways (i.e., TNF-α-mediated antiviral activity [136] etc.). Meanwhile, the appropriate combinations of host-targeted antivirals may improve efficacy, expand antiviral activity, and reduce the likelihood of drug resistance. For instance, McHutchison et al. reported that in HCV-infected patients who failed to respond to treatment with peginterferon alfa and Ribavirin previously, re-treatment with Telaprevir combined with peginterferon alfa and Ribavirin was proven to be more effective than re-treatment with either drug alone [137,138].

Although host-targeting antiviral drug discovery strategies have become a major focus of research, very few human host proteins were explored as antiviral targets. In addition, there are challenges in developing host-directed therapies. For example, emerging evidence suggests that viral resistance against host antivirals does occur [136]. Meanwhile, host-targeted therapies may produce some expected or unexpected side effects, since they may target host cell functions that are critical for cell survival [139]. Therefore, a more in-depth analysis of possible cytotoxicity or the potential to worsen the infection is needed when treating with host-targeted antivirals.

Genetic and evolutionary knowledge provides new insights into the pathogenesis of diseases by identifying specific genes or pathways associated with diseases, and thus provides opportunities to discover new drug targets. By summarizing the properties of 36 approved antiviral targets in genetics, cell biology and evolutionary biology, we found that the targets feature certain characteristics: (1) median genetic score(s) associated with viral disease(s) equal to or greater than three; (2) subcellular localization on the cell membrane; (3) origins in the Eumetazoan period. Based on these three principles, we used the gene set from the SCG-Drug database to screen new antiviral targets and obtained 35 host targets. By analyzing these candidate targets, we predicted some potential repositioned drugs and identified some new targets that may be used for de novo drug discovery.

Through the continuous development of biotechnology, significant advances have been made in high-throughput sequencing, omics analysis techniques, and molecular biology. Consequently, an increasing number of genetic, subcellular location, and evolutionary information about various genes will be accumulated. Especially in the context of the COVID-19 pandemic, the rapid screening of drug targets based on the above information will facilitate the discovery of druggable antiviral host targets, so as to promote the development of antiviral drugs.

## Figures and Tables

**Figure 1 viruses-13-02117-f001:**
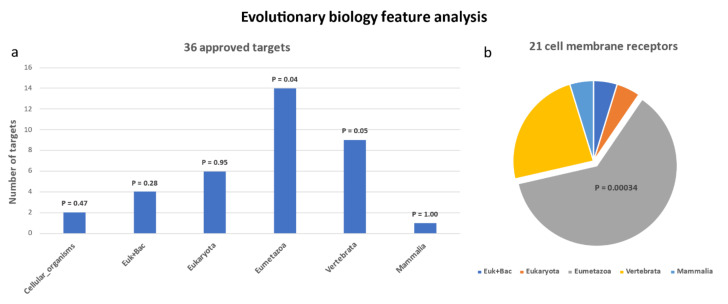
The statistical histogram of evolutionary features of approved antiviral receptors and cell membrane receptors. The results of the evolutionary information statistics showed that the 36 approved targets were distributed mainly in genes belonging to class 6 (Eumetazoa) (14/36, *p* = 0.04, hypergeometric assay) (**a**). As for the 21 cell membrane receptors, 13 of them were from the Eumetazoan period, and the tendency for enrichment was more pronounced (*p* = 0.00034, hypergeometric test) (**b**).

**Figure 2 viruses-13-02117-f002:**
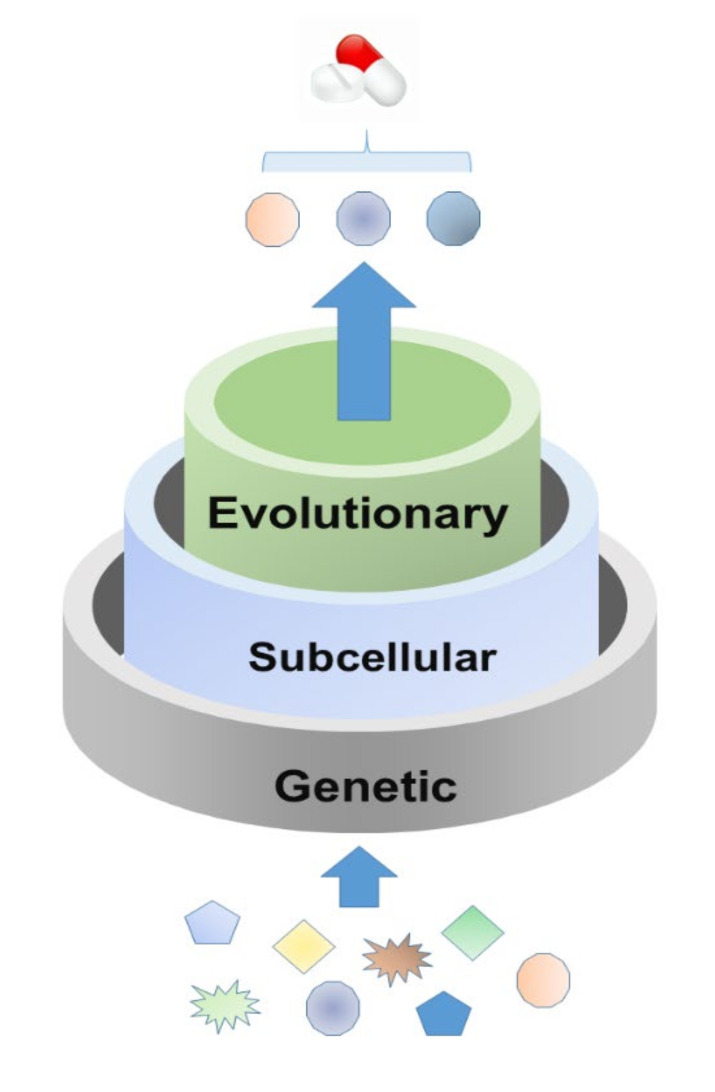
Antiviral host targets screening by genetic, subcellular location, and evolutionary biology features (different shapes and colors of symbols represent various genes related to viral diseases). Based on the analysis of the genetic features, subcellular localization, and evolutionary features of the approved antiviral host targets, it is summarized that an effective antiviral target generally exhibits the following three characteristics: (1) a median of genetic scores associated with viral disease (s) equal to or greater than three; (2) subcellular localization on the cell membrane; (3) origins in the Eumetazoan period.

**Figure 3 viruses-13-02117-f003:**
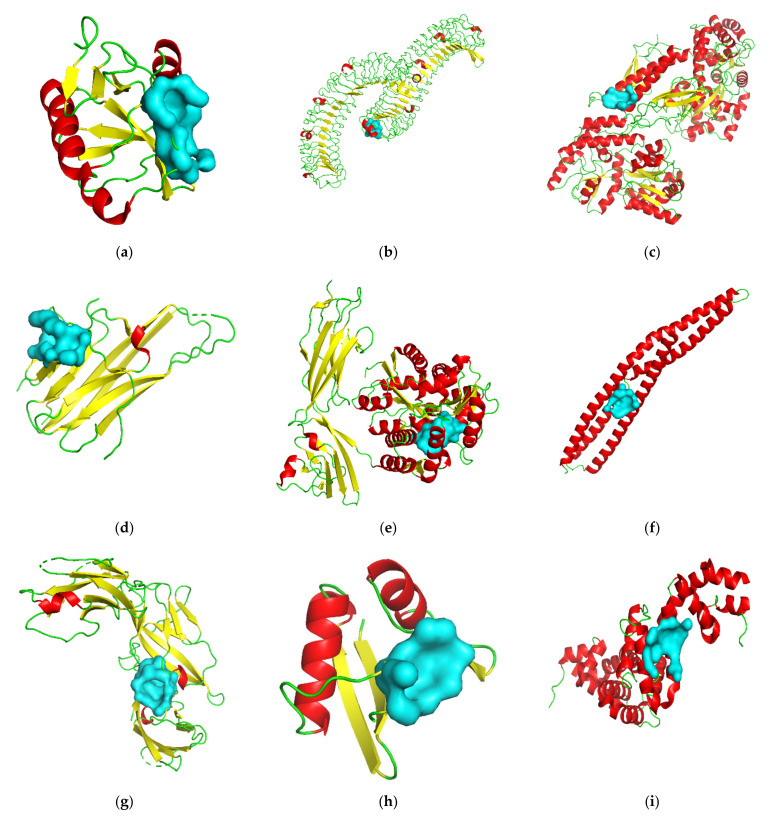
The best druggable pocket (shown in cyan surface mode) of each target predicted by PockDrug. A total of 11 candidate genes with known protein crystal structures and with one or more druggable pockets are presented, including *CD209* (**a**), *LGR5* (**b**), *CRHR2* (**c**), *FAS* (**d**), *AGER* (**e**), *BIN1* (**f**), *IL1RL1* (**g**), *PARD6A* (**h**), *RGS7* (**i**), *RHOU* (**j**) and *TNFSF10* (**k**). All the predicted information by PockDrug is summarized in Table 3 and Appendix A. Pockets with druggability scores of 0.5 or more were considered druggable.

**Table 1 viruses-13-02117-t001:** Basic information of approved antiviral host targets.

Gene Symbol	Uniprot ID	Protein Name	SCG Virus Diseases Median Score	Subcellular Location	Origin	Associated Virus Diseases Name (Only List the Top 3) *
*C1QA*	P02745	Complement C1q subcomponent subunit A	1	Secreted	Vertebrata	Herpes Simplex
*C1QB*	P02746	Complement C1q subcomponent subunit B	1	Secreted	Vertebrata	Herpes Simplex; Carcinoma, Merkel Cell
*C1QC*	P02747	Complement C1q subcomponent subunit C	1	Secreted	Vertebrata	Herpes Simplex; Adenoviridae Infections
*C1R*	P00736	Complement C1r subcomponent, EC 3.4.21.41	3	Secreted	Vertebrata	AIDS Dementia Complex
** *CCR5* **	**P51681**	**C-C chemokine receptor type 5, C-C CKR-5, CC-CKR-5, CCR-5, CCR5**	**3**	**Cell** **membrane**	**Eumetazoa**	**Hepatitis, Viral, Human; West Nile Fever; Hepatitis C**
** *CD4* **	**P01730**	**T-cell surface glycoprotein CD4**	**3**	**Cell** **membrane**	**Vertebrata**	**Measles; Picornaviridae Infections; HIV Infections**
** *CHRNA3* **	**P32297**	**Neuronal acetylcholine receptor subunit alpha-3**	**3**	**Cell** **membrane**	**Eumetazoa**	**AIDS Dementia Complex; Carcinoma, Merkel Cell; Influenza, Human**
** *CHRNA4* **	**P43681**	**Neuronal acetylcholine receptor subunit alpha-4**	**1**	**Cell** **membrane**	**Eumetazoa**	**AIDS Dementia Complex; Influenza, Human; Picornaviridae Infections**
** *CHRNA7* **	**P36544**	**Neuronal acetylcholine receptor subunit alpha-7**	**3**	**Cell** **membrane**	**Eumetazoa**	**AIDS Dementia Complex**
** *CXCR4* **	**P61073**	**C-X-C chemokine receptor type 4**	**3**	**Cell** **membrane**	**Eumetazoa**	**Herpes Simplex; Papillomavirus Infections; Epstein-Barr Virus Infections**
** *DRD2* **	**P14416**	**D(2) dopamine receptor**	**1**	**Cell** **membrane**	**Eumetazoa**	**AIDS Dementia Complex; Hemorrhagic Fevers, Viral; Hepatitis**
** *ENPP1* **	**P22413**	**Ectonucleotide pyrphosphatase/phosphodiesterase family member 1**	**3**	**Cell** **membrane**	**Eukaryota**	**West Nile Fever; Hepatitis; AIDS Dementia Complex**
** *FCGR1A* **	**P12314**	**High affinity immunoglobulin gamma Fc receptor I**	**1**	**Cell** **membrane**	**Eumetazoa**	**Leukoencephalopathy, Progressive Multifocal; Hemorrhagic Fever, Ebola; Hemorrhagic Fevers, Viral**
** *FCGR2A* **	**P12318**	**Low affinity immunoglobulin gamma Fc region receptor II-a**	**2**	**Cell** **membrane**	**Eumetazoa**	**Leukoencephalopathy, Progressive Multifocal; West Nile Fever; Picornaviridae Infections**
** *FCGR2B* **	**P31994**	**Low affinity immunoglobulin gamma Fc region receptor II-b**	**1**	**Cell** **membrane**	**Eumetazoa**	**Picornaviridae Infections; Hemorrhagic Fevers, Viral; Hepatitis C**
** *FCGR2C* **	**P31995**	**Low affinity immunoglobulin gamma Fc region receptor II-c**	**1**	**Cell** **membrane**	**Eumetazoa**	**HIV Infections; Herpes Simplex;** **Hepatitis C**
** *FCGR3A* **	**P08637**	**Low affinity immunoglobulin gamma Fc region receptor III-A**	**3**	**Cell** **membrane**	**Eumetazoa**	**Herpesviridae Infections; Hepatitis, Viral, Human; Measles**
** *FCGR3B* **	**O75015**	**Low affinity immunoglobulin gamma Fc region receptor III-B**	**1**	**Cell** **membrane**	**Eumetazoa**	**Influenza, Human; Picornaviridae Infections; West Nile Fever**
** *GRIN3A* **	**Q8TCU5**	**Glutamate receptor ionotropic, NMDA 3A, GluN3A**	**3**	**Cell** **membrane**	**Vertebrata**	**AIDS Dementia Complex**
** *HLA-B* **	**P18465**	**HLA class I histocompatibility antigen, B alpha chain**	**3**	**Cell** **membrane**	**Vertebrata**	**HIV Infections; Hepatitis; Picornaviridae Infections**
** *IFNAR1* **	**P17181**	**Interferon alpha/beta receptor 1**	**1**	**Cell** **membrane**	**Vertebrata**	**Leukoencephalopathy, Progressive Multifocal; Hemorrhagic Fevers, Viral; West Nile Fever**
** *IFNAR2* **	**P48551**	**Interferon alpha/beta receptor 2**	**3**	**Cell** **membrane**	**Vertebrata**	**Hepatitis; Picornaviridae Infections; Leukoencephalopathy, Progressive Multifocal**
*IMPDH1*	P20839	**Inosine-5’-monophosphate dehydrogenase 1**	**——**	Cytoplasm	Cellular_ organisms	**——**
*IMPDH2*	P12268	Inosine-5’-monophosphate dehydrogenase 2	1	Nucleus	Cellular_ organisms	Sarcoma, Kaposi
** *NEU1* **	**Q99519**	**N-acetyl-alpha-neuraminidase 1**	**2**	**Cell** **membrane**	**Euk + Bac**	**Carcinoma, Merkel Cell; Leukoencephalopathy, Progressive Multifocal; Sarcoma, Kaposi**
*NEU2*	Q9Y3R4	**N-acetyl-alpha-neuraminidase 2**	**——**	Cytoplasm	Euk + Bac	**——**
*NR1I2*	O75469	Nuclear receptor subfamily 1 group I member 2	1	Nucleus	Eumetazoa	Hepatitis; Measles; Paramyxoviridae Infections…
*NT5C2*	P49902	Cytosolic purine 5’-nucleotidase, EC 3.1.3.5	1	Cytoplasm	Eukaryota	AIDS Dementia Complex; Cytomegalovirus Infections; Sarcoma, Kaposi
*PNP*	P00491	Purine nucleoside phosphorylase, PNP, EC 2.4.2.1	1	Cytoplasm	Euk + Bac	Cytomegalovirus Infections; AIDS Dementia Complex; Epstein-Barr Virus Infections
** *SLCO1B1* **	**Q9Y6L6**	**Solute carrier organic anion transporter family member 1B1**	**3**	**Cell membrane**	**Mammalia**	**Fatigue Syndrome, Chronic; Picornaviridae Infections; HIV Infections**
** *SLCO2B1* **	**O94956**	**Solute carrier organic anion transporter family member 2B1**	**——**	**Cell membrane**	**Eumetazoa**	**——**
*TERT*	O14746	Telomerase reverse transcriptase, EC 2.7.7.49	1	Nucleolus	Eukaryota	Carcinoma, Merkel Cell; Leukoencephalopathy, Progressive Multifocal; Influenza, Human
*TOP2A*	P11388	DNA topoisomerase 2-alpha, EC 5.6.2.2	1	Nucleoplasm	Eukaryota	Leukoencephalopathy, Progressive Multifocal; Carcinoma, Merkel Cell; Hepatitis, Viral, Human
*TUBA4A*	P68366	Tubulin alpha-4A chain	4	Cytoskeleton	Eukaryota	AIDS Dementia Complex; Leukoencephalopathy, Progressive Multifocal
*TUBB*	P07437	Tubulin beta chain	——	Cytoplasm, cytoskeleton	Eukaryota	——
*TYMS*	P04818	Thymidylate synthase, TS, TSase, EC 2.1.1.45	1	Nucleus	Euk + Bac	Sarcoma, Kaposi; Influenza, Human; Hepatitis

Genes located on the cell membrane are shown in bold. * Complete candidate targets and diseases association information are listed in Appendix A. Disease names are listed in descending order of SCG scores.

**Table 2 viruses-13-02117-t002:** Basic information of candidate targets.

Gene Symbol	Uniprot ID	Protein Name	Drug Information	Recognized Target	SCG Virus Disease(S) Median Score	Associated Virus Disease(s) Name (Only List the Top 3) *	Evolutionary Age	Pubmed Number ^a^
*ADRA2A*	P08913	Alpha-2A adrenergic receptor	known	Yes	3	AIDS Dementia Complex; Hepatitis	Eumetazoa	13
*ADRB2*	P07550	Beta-2 adrenergic receptor	known	Yes	3	Leukoencephalopathy, Progressive Multifocal; Paramyxoviridae Infections; Picornaviridae Infections	Eumetazoa	52
*ADRB3*	P13945	Beta-3 adrenergic receptor	known	Yes	3	Hepatitis; AIDS Dementia Complex; Picornaviridae Infections	Eumetazoa	1
*AGER*	Q15109	Advanced glycosylation end product-specific receptor	unknown	No	3	Leukoencephalopathy, Progressive Multifocal; Hepatitis; AIDS Dementia Complex	Eumetazoa	27
*AGTR1*	P30556	Type-1 angiotensin II receptor	known	Yes	3	Leukoencephalopathy, Progressive Multifocal; Paramyxoviridae Infections; Hepatitis	Eumetazoa	3
*ASIC1*	P78348	Acid-sensing ion channel 1, ASIC1	known	Yes	3	Leukoencephalopathy, Progressive Multifocal	Eumetazoa	3
*BIN1*	O00499	Myc box-dependent-interacting protein 1	unknown	No	6	Fatigue Syndrome, Chronic; AIDS Dementia Complex	Eumetazoa	14
*BSG*	P35613	Basigin	unknown	No	3	Hepatitis; West Nile Fever; Epstein–Barr Virus Infections	Eumetazoa	19
*CALHM1*	Q8IU99	Calcium homeostasis modulator protein 1	unknown	No	9	AIDS Dementia Complex	Eumetazoa	1
*CD209*	Q9NNX6	CD209 antigen	unknown	Yes	3	West Nile Fever; Hepatitis; Hemorrhagic Fever, Ebola	Eumetazoa	65
*CD86*	P42081	T-lymphocyte activation antigen CD86	known	Yes	3	Respiratory Syncytial Virus Infections; Picornaviridae Infections; Hepatitis D	Eumetazoa	758
*CRHR2*	Q13324	Corticotropin-releasing factor receptor 2	unknown	Yes	3	Hepatitis; Fatigue Syndrome, Chronic; West Nile Fever	Eumetazoa	4
*DRD4*	P21917	D(4) dopamine receptor	known	Yes	3	AIDS Dementia Complex	Eumetazoa	5
*DRP2*	Q13474	Dystrophin-related protein 2	unknown	No	3	AIDS Dementia Complex	Eumetazoa	1
*FAS*	P25445	Tumor necrosis factor receptor superfamily member 6	unknown	Yes	3	Picornaviridae Infections; AIDS Dementia Complex; Carcinoma, Merkel Cell	Eumetazoa	1588
*GHSR*	Q92847	Growth hormone secretagogue receptor type 1	known	Yes	3	Carcinoma, Merkel Cell	Eumetazoa	7
*GPC5*	P78333	Glypican-5	unknown	No	3	Leukoencephalopathy, Progressive Multifocal;	Eumetazoa	1
*GPR65*	Q8IYL9	Psychosine receptor	unknown	No	6	Leukoencephalopathy, Progressive Multifocal	Eumetazoa	1
*GRM4*	Q14833	Metabotropic glutamate receptor 4	known	Yes	3	Leukoencephalopathy, Progressive Multifocal; Hepatitis; Picornaviridae Infections	Eumetazoa	1
*GRPR*	P30550	Gastrin-releasing peptide receptor	unknown	Yes	3	AIDS Dementia Complex	Eumetazoa	1
*IL1R1*	P14778	Interleukin-1 receptor type 1	known	Yes	3	Leukoencephalopathy, Progressive Multifocal; Picornaviridae Infections; HIV Infections	Eumetazoa	17
*IL1RL1*	Q01638	Interleukin-1 receptor-like 1	unknown	No	3	Influenza, Human; Hepatitis D; West Nile Fever	Eumetazoa	31
*KCNQ1*	P51787	Potassium voltage-gated channel subfamily KQT member 1	known	Yes	3	Picornaviridae Infections; AIDS Dementia Complex; Measles	Eumetazoa	5
*LGR5*	O75473	Leucine-rich repeat-containing G-protein coupled receptor 5	unknown	Yes	3	HIV Infections; Influenza, Human; Paramyxoviridae Infections	Eumetazoa	25
*MAG*	P20916	Myelin-associated glycoprotein	unknown	No	3	Leukoencephalopathy, Progressive Multifocal	Eumetazoa	100
*NPSR1*	Q6W5P4	Neuropeptide S receptor	known	Yes	3	Hepatitis; Respiratory Syncytial Virus Infections; Influenza, Human	Eumetazoa	1
*PARD6A*	Q9NPB6	Partitioning defective 6 homolog alpha	unknown	No	3	AIDS Dementia Complex	Eumetazoa	2
*PERP*	Q96FX8	p53 apoptosis effector related to PMP-22	unknown	No	3	Carcinoma, Merkel Cell; Leukoencephalopathy, Progressive Multifocal	Eumetazoa	5
*RGS7*	P49802	Regulator of G-protein signaling 7	unknown	No	3	Leukoencephalopathy, Progressive Multifocal	Eumetazoa	1
*RHOU*	Q7L0Q8	Rho-related GTP-binding protein RhoU	unknown	No	3	Carcinoma, Merkel Cell	Eumetazoa	4
*SGCA*	Q16586	Alpha-sarcoglycan, Alpha-SG	unknown	No	14	Fatigue Syndrome, Chronic	Eumetazoa	7
*SHB*	Q15464	SH2 domain-containing adapter protein B	unknown	No	3	AIDS Dementia Complex	Eumetazoa	19
*SYNE2*	Q8WXH0	Nesprin-2	unknown	No	7	Fatigue Syndrome, Chronic;	Eumetazoa	4
*TGFBR1*	P36897	TGF-beta receptor type-1	known	Yes	3	Hepatitis; Hepatitis B; Carcinoma, Merkel Cell	Eumetazoa	29
*TNFSF10*	P50591	Tumor necrosis factor ligand superfamily member 10	unknown	No	3	Hepatitis D; HIV Infections; Leukoencephalopathy, Progressive Multifocal	Eumetazoa	224

^a^ Refers to the number of NCBI Pubmed (https://www.ncbi.nlm.nih.gov/, accessed on 20 October 2021) studies related to this gene and virus. * Complete candidate targets and diseases association information are listed in Appendix A. Disease names are listed in descending order of SCG scores.

**Table 3 viruses-13-02117-t003:** Druggability information of candidate targets.

Gene Symbol	PDB ID	Protein Name	Resolution (Å)	Ligand(s) ^a^	Number of Pockets	Number of Druggable Pockets (Score ≥ 0.5)	Best Druggable Pocket Score
*AGER*	3O3U	Advanced glycosylation end product-specific receptor	1.50	MLR, SO4	24	13	1.00
*BIN1*	2FIC	Myc box-dependent-interacting protein 1	1.99	XE	11	7	0.91
*BSG*	3I84	Basigin	2.00	CL	2	0	——
*CD209*	2XR6	CD209 antigen	1.35	07B, MAN, AE9, CA, CL	2	2	0.78
*CRHR2*	3N93-AB	Corticotropin-releasing factor receptor 2	2.50	MAL, GOL	41	18	1.00
*FAS*	3TJE-F	Tumor necrosis factor receptor superfamily member 6	1.93	CD, EDO, CL	4	1	0.84
*IL1RL1*	4KC3-B	Interleukin-1 receptor-like 1	3.27	NAG, MSE	11	7	0.98
*LGR5*	4UFR-AC	Leucine-rich repeat-containing G-protein coupled receptor 5	2.20	NAG, CL	34	21	0.99
*PARD6A*	1WMH-B	Partitioning defective 6 homolog alpha	1.50	——	4	3	0.99
*RGS7*	2A72	Regulator of G-protein signaling 7	2.00	CL	7	2	0.60
*RHOU*	2Q3H	Rho-related GTP-binding protein RhoU	1.73	GDP, MG	6	1	0.93
*TNFSF10*	1DG6-A	Tumor necrosis factor ligand superfamily member 10	1.30	ZN, CL	2	1	0.65

^a^ The ligand information is derived from the RCSB PDB (https://www.rcsb.org/, accessed on 20 October 2021) [105].

## Data Availability

Publicly available datasets were analyzed in this study. This data can be found here: DrugBank (https://go.drugbank.com/, accessed on 20 October 2021) and the SCG-Drug database (Quan et al. 2019).

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
