# Peer review of "Facilitating Antiviral Drug Discovery Using Genetic and Evolutionary Knowledge"

_viruses, 2021, doi:10.3390/v13112117_

Round 1
Reviewer 1 Report
Review on ‘Facilitating Antiviral Drug Discovery by Genetic and Evolutionary Knowledge’, by Xu et al., 2021.
This paper reviews a strategy for host-targeted antiviral discovery through filtering via genetic and evolutionary features. The manuscript reviews the genetic, subcellular and evolutionary characteristics of human host targets for antivirals. Finally, the authors summarize novel druggable antiviral targets of a number of host proteins. Apart from some grammatical errors, the manuscript is well written and gives a comprehensive review of a narrow field of antivirals. The manuscript identifies several host genes already identified as targets for antiviral therapies as well as some novel potential targets.
Some specific comments:
Line 35: Coronavirus is one word and the authors should mention SARS-CoV-2 here as well as COVID-19, as SARS-CoV-2 is the etiological agent responsible for the pandemic.
Line 46: Please clarify or reword “easy outbreak”, this is not clear to the reader.
Line 174: There appears to be a typographic error in the word ‘domencuted’ which I assume should be documented.
Line 206: I assume ‘Uniport’ should be Uniprot.
Line 220: The sentence should be changed to read “…viruses initially interact with the host…” as the there are many interactions with the host throughout the viral lifecycle, not just attachment/entry.
Line 227: There appears to be a typographic error, this should change to plural ‘viruses’, or the previous line 226 should have a/the before ‘virus’.
Line 281: I think generate should change to ‘identify’ as it’s not technically correct, these genes are identified, not generated.
Line 307: “For 4,937 genes that has no..” should change to “For 4,937 genes that have no…” to be grammatically correct.
General comments:
- It would be good if the authors could comment on identification of genes determined as causal of severe viral disease by Mendelian Randomisation which is different to and more rigorous than genome-wide association studies (GWAS) which only identifies important disease genes only by association to a phenotype. For COVID-19, examples of these causal genes include JAK1, JAK2, TNF, IL6, TMPRSS2 as well as some of the same genes the authors list as associated (i.e., ILIR1) and could be discussed in light of the presented findings. I encourage authors to look at the recent study by Pairo-Castineira et al., 2020 (https://www.nature.com/articles/s41586-020-03065-y).
- While there are multiple advantages to host-targeted antivirals, such as the higher barrier to resistance, the disadvantages should also be mentioned to ensure a balanced academic argument. For example, any approach targeting host cell functions involved in the viral lifecycle can lead to expected or unexpected side effects because they target host cellular functions that may be essential to cell survival, they can be toxic. Also, host-targeted antivirals may be impacted by host genetic polymorphisms that could alter their ability to block their target protein function. In clinical settings suboptimal responses have been reported to alisporivir in 10-15% of patients and these potential effects of SNPs etc. should be discussed. The impacts of polymorphisms are discussed for HBV vaccine effectiveness and influenza susceptibility, but not for host-targeted antivirals.
- While the manuscript clearly focuses on subcellular localization, it would be worth mentioning that there are multiple avenues for host-targeted antivirals including numerous innate immune targets (i.e., TLR agonists etc.).
- While most direct-acting antivirals do have a lower barrier to resistance, they are almost always used to combinations to overcome this issue and as demonstrated by HCV sustained virological responses, works extremely well and should be acknowledged.
Author Response
We sincerely thank the reviewer for reading our paper carefully and giving the above positive comments. Your suggestions are very helpful for us to improve our work. We are very sorry for the mistakes in this manuscript and inconvenience they caused during your reading. The revised manuscript has been thoroughly checked and the grammatical errors and typos have been corrected. We have also revised the manuscript according to your general comments (presented by using the “track changes” mode in MS Word) and submit it to the journal again.
The point-to-point responses to reviewers’ questions are listed as below.
- Line 35: Coronavirus is one word and the authors should mention SARS-CoV-2 here as well as COVID-19, as SARS-CoV-2 is the etiological agent responsible for the pandemic.
Reply:
Thanks to the reviewer for reminding us, we apologize for the mistakes, and we have corrected them as ‘Coronavirus’ and reorganized the statements according to your professional suggestions. (Line 35-36 of page 1)
- Line 46: Please clarify or reword “easy outbreak”, this is not clear to the reader.
Reply:
We thank the reviewer for pointing out this issue. We have replaced this word with "wide spread" to avoid possible misunderstandings. (Line 46 of page 2)
- Line 174: There appears to be a typographic error in the word ‘domencuted’ which I assume should be documented.
Reply:
Thanks to the reviewer for reminding us, we apologize for the mistakes, and we have corrected it as ‘documented’. (Line 188 of page 4)
- Line 206: I assume ‘Uniport’ should be Uniprot.
Reply:
Thank you for pointing out this problem in our manuscript. We have fixed the error. (Line 190 of page 4, Line 220 of page 6)
- Line 220: The sentence should be changed to read “…viruses initially interact with the host…” as the there are many interactions with the host throughout the viral lifecycle, not just attachment/entry.
Reply:
Thanks for your constructive suggestion. We agree and have updated the sentence. (Line 234 of page 6)
- Line 227: There appears to be a typographic error, this should change to plural ‘viruses’, or the previous line 226 should have a/the before ‘virus’.
Reply:
Thank you for pointing out this problem. We’ve changed the word ‘virus’ to ‘viruses’ accordingly. (Line 241 of page 7)
- Line 281: I think generate should change to ‘identify’ as it’s not technically correct, these genes are identified, not generated.
Reply:
Thank for your comments. We’ve replaced the word ‘generate’ with ‘identify’. (Line 299 of page 8)
- Line 307: “For 4,937 genes that has no..” should change to “For 4,937 genes that haveno…” to be grammatically correct.
Reply:
Thank for your comments. We are very sorry for the grammatical mistakes.
We have revised the sentence based on your comments. (Line 328 of page 9)
General comments:
- It would be good if the authors could comment on identification of genes determined as causal of severe viral disease by Mendelian Randomisation which is different to and more rigorous than genome-wide association studies (GWAS) which only identifies important disease genes only by association to a phenotype. For COVID-19, examples of these causal genes include JAK1, JAK2, TNF, IL6, TMPRSS2 as well as some of the same genes the authors list as associated (i.e., ILIR1) and could be discussed in light of the presented findings. I encourage authors to look at the recent study by Pairo-Castineira et al., 2020 (https://www.nature.com/articles/s41586-020-03065-y).
Reply:
Thanks for your suggestion. We strongly agree with you that MR is indeed an important analysis method. We have already cited this article in the introduction section for a brief comment in the previous manuscript. We have now added a specific description of this method (Line 94-101 of page 2-3, ref [34], highlighted in yellow). Also, in section 2.1, some comments on the available results of the MR method have been included to further emphasize the importance and wide application of this method. (Line 161-167 of page 4)
- While there are multiple advantages to host-targeted antivirals, such as the higher barrier to resistance, the disadvantages should also be mentioned to ensure a balanced academic argument. For example, any approach targeting host cell functions involved in the viral lifecycle can lead to expected or unexpected side effects because they target host cellular functions that may be essential to cell survival, they can be toxic. Also, host-targeted antivirals may be impacted by host genetic polymorphisms that could alter their ability to block their target protein function. In clinical settings suboptimal responses have been reported to alisporivir in 10-15% of patients and these potential effects of SNPs etc. should be discussed. The impacts of polymorphisms are discussed for HBV vaccine effectiveness and influenza susceptibility, but not for host-targeted antivirals.
Reply:
Thank you for your constructive suggestions, which is valuable for improving the accuracy of the manuscript. More discussions about the disadvantages of host-targeted antivirals have been included in the Discussion and Conclusion section in the revised manuscript. (Line 536-542 of page 15) Also, based on your professional comments, the relevant content and literature about the influence of genetic polymorphisms on the efficacy of host-targeted antivirals have been incorporated in section 2.1. Genetic features. (Line 139-143 of page 3)
- While the manuscript clearly focuses on subcellular localization, it would be worth mentioning that there are multiple avenues for host-targeted antivirals including numerous innate immune targets (i.e., TLR agonists etc.).
Reply:
Thank you for the above suggestion. In the Discussion and Conclusion section, we have provided some additional examples to illustrate the multiple pathways of host-targeted antiviral treatment strategies (Line 524-528 of page 16).
- While most direct-acting antivirals do have a lower barrier to resistance, they are almost always used to combinations to overcome this issue and as demonstrated by HCV sustained virological responses, works extremely well and should be acknowledged.
Reply:
Thanks for your kind suggestions. In the Discussion and Conclusion section, we have included some particularly outstanding works to illustrate the effective use of combinatorial strategies for host-targeted antivirals. (Line 528-534 of page 16-17)
Thanks again to the reviewer on suggesting how to further improve this manuscript. We have studied comments carefully and have made corresponding corrections which we hope can meet with approval.
Reviewer 2 Report
- Could you please expand legends for Fig. 1-3.
- Could you please provide protein names instead of IDs in Tables 1-3.
- Could you please describe drug combination strategies.
Author Response
Sincere thanks should be given to the reviewer for the constructive comments and suggestions. We have tried our best to revise the manuscript according to your kind and construction comments and suggestions (presented by using the “track changes” mode in MS Word) and submit it to the journal again.
The point-to-point responses to reviewers’ questions are listed as below.
1. Could you please expand legends for Fig. 1-3.
Reply:
We are grateful for the reviewer’s suggestions, and we have re-written the manuscript with more detailed description about Fig.1-3. (Line 291-294 of page 8, Line 311-314 of page 9, Line 508-511 of page 16)
2. Could you please provide protein names instead of IDs in Tables 1-3.
Reply:
Thank the reviewer for the constructive suggestions. We have revised the Tables 1-3 to address your concerns and hope that it is now clearer. (Line 190 of page 4, Line 338 of page 10, Line 367 of page 12)
3. Could you please describe drug combination strategies.
Reply:
We appreciate the reviewer’s insightful suggestion and agree that it would be useful to demonstrate drug combination strategies. In the Discussion and Conclusion section, we have included some particularly outstanding works to illustrate the effective use of combinatorial strategies for host-targeted antivirals. (Line 528-534 of page 16-17)
Thanks again to the reviewer on suggesting how to further improve this manuscript. We have studied comments carefully and have made corresponding corrections which we hope can meet with approval.
Round 2
Reviewer 2 Report
The ms was revised appropriately.